# Differential Inhibition of Target Gene Expression by Human microRNAs

**DOI:** 10.3390/cells8080791

**Published:** 2019-07-30

**Authors:** Peng Li, Yi Chen, Conslata Awino Juma, Chengyong Yang, Jinfeng Huang, Xiaoxiao Zhang, Yan Zeng

**Affiliations:** Department of Zoology, College of Life Sciences, Nanjing Agricultural University, Nanjing, Jiangsu 210095, China

**Keywords:** miRNA, miRNA target gene, target gene prediction, differential gene suppression, differential gene expression

## Abstract

microRNAs (miRNAs) exert their functions by repressing the expression of their target genes, but most miRNA target genes are unknown, and the degree to which a miRNA differentially inhibits the expression of its targets is underappreciated. We selected human miR-1, miR-122, and miR-124 as representatives to investigate the reliability of miRNA target predictions and examine how miRNAs suppress their targets. We constructed miRNA target gene reporter libraries based on prediction programs TargetScan, miRanda, and PicTar, and performed large-scale reporter assays to directly evaluate whether and how strongly a predicted target gene is repressed by its miRNA. We then performed statistical analyses to examine parameters that contributed to the miRNA inhibition of target genes. We found that the three programs have approximately 72–85% success rates in predicting genuine targets and that the miRNA inhibition of different targets varies in extent. We also identified parameters that could predict the degrees of miRNA repression, and further showed that differential miR-124 repression might contribute to differential gene expression in vivo. Our studies systematically investigated hundreds of miRNA target genes, shed light on factors influencing miRNA functions, and suggested a new mechanism by which differential target repression by miRNAs regulates endogenous gene expression.

## 1. Introduction

microRNAs (miRNAs) are a family of small, non-coding RNAs that exert important biological functions by inhibiting the expression of their target genes [1,2]. A miRNA, in complex with an Argonaute protein, binds to its target mRNA, usually at the 3′ untranslated region (UTR), via base pairing. If the sequence complementarities are perfect or nearly perfect, the Argonaute protein can cleave the mRNA. In animal cells, however, the degree of matches between miRNAs and most target mRNAs is low, so the Argonaute protein does not cleave the mRNAs; instead, it recruits proteins, such as TNRC6, to destabilize the mRNAs and/or block translation [3]. Consequently, miRNA action typically reduces both the mRNA and protein levels of target genes [2,4].

miRNAs are approximately 22 nucleotides (nt) long, but their sequences do not contribute to target recognition uniformly, with nt 2–8 from the 5′ end, the so-called seed region, being the most discriminatory. An animal mRNA generally needs to pair perfectly to only the seed or one of its slight variations to be selected and suppressed by a miRNA, although a minority of targets lack a canonical seed match and/or bind critically to the central or 3′ part of miRNAs [5,6,7,8,9,10,11,12,13]. Such discoveries have allowed the development of numerous algorithms to predict genome-wide miRNA target genes in model organisms, including humans [2]. While they all compare the sequences of miRNAs and mRNAs, these models differ in the ways they choose the sequence databases, and evaluate and score the relative importance of seed matches, the conservation of miRNA binding sites or miRNA response elements (MREs), the binding energy of miRNA-mRNA duplexes, and other sequence and structural features of the 3′ UTRs. As a result, there is variable overlap between target genes predicted by different programs. Nonetheless, they all point to hundreds or thousands of target genes for a typical human miRNA, supporting the notion that miRNAs repress the expression of a large number of target genes and, hence, play pervasive regulatory roles in vivo [10].

Once predicted, candidate miRNA target genes need to be confirmed experimentally, which is necessary to understand the mechanisms and functions of miRNAs. There are three general strategies employed to verify target genes. The first is to biochemically isolate Argonaute:miRNA:mRNA complexes and then profile the mRNAs [14,15,16,17,18]. This setup interrogates the physical interactions between miRNAs and mRNAs and has demonstrated a vast number of potential miRNA binding sites in vivo, yet binding by itself does not equate to repression by miRNAs [12,19]. The second, termed the profiling strategy here, is to overexpress, knockdown, or knockout a miRNA, and then examine global mRNA expression by microarray or RNA-sequencing (RNA-seq) techniques [20,21]. If, for example, a miRNA is overexpressed, then the levels of its target mRNAs should be reduced. This approach is conceptually simple, high-throughput, and has become the favorite method to investigate miRNA target genes, especially at a large scale. Indeed, most of our current understanding of the parameters governing miRNA:mRNA interactions, such as seed categories and target AU content, is derived from results of such studies [9,12]. The profiling strategy, however, cannot ascertain that changes in mRNA levels, or the degrees of such changes, are directly or solely due to the gain or loss of a miRNA in question. The third, reporter strategy, is to insert an mRNA sequence containing the putative MRE(s) into the 3′ UTR of a reporter gene, and then examine whether its expression is subject to miRNA inhibition [22]. Although performed in an artificial setting, this method provides the most stringent test of whether, and the extent to which, a miRNA can directly inhibit the expression of a putative target gene. Nevertheless, it requires cloning and analyzing genes individually and hence, is cumbersome. As a result, this approach has been confined to verifying a small number of specific targets after they have been suggested by genetics, computation, or other methods [22,23,24,25]. Slutskin et al. [26] employed a massively parallel report assay to investigate miRNA:target interactions, but their targets consisted of artificially designed sequences.

The sheer size of the potential target pools, along with the advantages and disadvantages of the three aforementioned strategies, present a challenge to the identification of genuine miRNA target genes. The vast majority of currently curated miRNA targets, e.g., by miRTarBase [27], are based on gene profiling evidence, which is obviously indirect and relatively weak. Consequently, most miRNA targets are considered unknown or unconfirmed. Another critical question is, how do miRNAs affect their numerous targets? Arvey et al. [28] and Garcia et al. [29] compared the ways in which different miRNAs suppress their respective targets, and Erhard et al. [30] and Nam et al. [31] investigated how a miRNA may differentially regulate the same genes in different cellular contexts. That a miRNA’s repressive activity varies among targets, or miRNA target efficacy, has been investigated using the profiling strategy [9,12,23,24,30]. As noted above, however, changes in mRNA levels and the degrees of changes cannot be unequivocally or completely contributed to miRNA manipulation, and how much individual targets are directly inhibited by miRNAs and endogenous gene expression without miRNA perturbation has not been examined under the same light.

We designed this study to assess how reliable target predictions were, as well as how miRNAs repressed their target genes, using the reporter strategy as an alternative to the commonly used profiling strategy. For representatives, we selected human miR-1, miR-122, and miR-124. miR-1 is a muscle-specific miRNA important for heart development and functions [32]. miR-122 is a liver-specific miRNA implicated in lipid metabolism and hepatitis C virus replication [33,34,35]. miR-124 is enriched in neurons and plays numerous roles in the nervous system, such as during neurogenesis [36]. These three miRNAs are relatively well-known, and a number of studies have examined their target genes, which allows us to compare results in this and future studies [9,12,14,19,21,24,25,29,31,32,33,34,35,36]. Our basic scheme was to construct libraries containing hundreds of firefly luciferase reporters based on target genes predicted by TargetScan, miRanda, and PicTar, among the earliest and most popular human miRNA target prediction programs [23,25,37], and then determine whether and how much miR-1, miR-122, or miR-124 repressed reporter expression in human cell cultures. This method has been used to estimate the success rates of target predictions at a much smaller scale or to test individual genes, but has never been applied systematically, as few studies have examined more than 10 targets per miRNA at a time [23,24,25,29]. Based on the wealth of newly acquired information, we then compared conclusions of the profiling strategy and reporter strategy and examined how various RNA features and program parameters predicted differential repression by miRNAs. For example, all programs calculate a score for every predicted gene, where the larger the absolute value is, the more likely the gene is to be a target, but how well do the scores really foretell miRNA target genes and how strongly is a target gene inhibited by a miRNA? Moreover, miR-1, miR-122, and miR-124 are highly expressed in muscles, the liver, and the nervous system, respectively, which provides an opportunity to examine endogenous mRNA expression in those tissues and correlate it to target inhibition by specific miRNAs.

## 2. Material and Methods

### 2.1. Target Selection

Tentative target genes of hsa-miR-1-3p, hsa-miR-122-5p, and hsa-miR-124-3p (miR-1, miR-122, and miR-124 for short, respectively) were downloaded from TargetScan (http://www.TargetScan.org/vert_70), miRanda (http://34.236.212.39/microrna/getGeneForm.do), and PicTar (http://pictar.mdc-berlin.de/) websites, from which a total of 194–196 targets were selected for each miRNA. Some selections were predicted by all three programs and identified using miRGator [38]. Also included were a few reported targets, according to miRTarBase 6.0 [27]; for miRTarBase entries, we considered only those with evidence of reporter assays, i.e., the most stringently tested. We then, as a general rule, designed primers to clone ~500 nt of a target mRNA with the predicted MRE in the middle, according to the longest annotated 3′ UTR in the UC Santa Cruz Genome Brower database, into a reporter plasmid. mRNA 3′ UTRs are highly variable in length, so 500 nt, presumably long enough to maintain the native structure of mRNAs, served as a compromise for overall consistency, which has also been applied in other studies [24,31]. Many mRNAs have more than one predicted MRE. We primarily examined mRNAs with a single MRE, except if all the predicted sites were closely clustered, e.g., within 300 nt, at which point we cloned a cDNA fragment with all the MREs flanked by ~240 nt extra sequences on each side to construct its reporter gene. Lastly, the whole range of ranking scores was represented in our collections so that the predictive values of these scores could be evaluated later. For TargetScan 7.0, we considered the cumulative weighted context ++ scores.

### 2.2. Cell Cultures

Human 293T, HeLa, and Huh7 cells were obtained from Sangon (Shanghai, China) and maintained at 37 °C and 5% CO_2_ in Dulbecco’s modified Eagle’s medium supplemented with 10% fetal bovine serum and 2 mM l-glutamine (Invitrogen, Carlsbad, CA, USA).

### 2.3. Molecular Cloning

Restriction enzymes, Taq polymerase, and T4 DNA Ligase were obtained from New England BioLabs (Ipswich, MA, USA). pRL-CMV, the control, Renilla luciferase-expressing plasmid, was from Promega (Madison, WI, USA). pCMV-luc and pSuper-7T have been previously described [39,40]. Oligonucleotide primers were ordered from Sangon. pSuper-gmiR-1, a human miR-1-expressing plasmid, was constructed by amplifying ~200-basepair-long genomic miR-1-2 sequences from human genomic DNA (Clontech, Mountain View, CA, USA) using primers 5′-GCAAGCTTCATCTGTTCATGACTAG-3′ and 5′-GCCTCGAGTTTTTACAGCTAACAACTTAG-3′, digesting the PCR product with HindIII and XhoI, and inserting it into pSuper-7T. pSuper-gmiR-122 was similarly constructed, with primers 5′-GCAAGCTTAGGTCACAATATGT-3′ and 5′-GCCTCGAGTGCAAAAGAGCCAG-3′, and pSuper-gmiR-124 with primers 5′-GCAAGCTTATTCCATCTTCTACCCA-3′ and 5′-GACTCGAGGGAGCTCCAACCCCT-3′. pCMV-luc-miR-1, a positive control reporter with 3′UTR sequences perfectly complementary to miR-1, was constructed by annealing 5′-CAGCTAGCATACATACTTCTTTACATTCCATACTCGAGCC-3′ and 5′-GGCTCGAGTATGGAATGTAAAGAAGTATGTATGCTAGCTG-3′, digesting the DNA with NheI and XhoI, and cloning it into pCMV-luc. pCMV-luc-miR-122 was likewise constructed by using primers 5′-CAGCTAGCAAACACCATTGTCACACTCCACAAACACCATTGTCACACTCCACTCGAGAC-3′ and 5′-GTCTCGAGTGGAGTGTGACAATGGTGTTTGTGGAGTGTGACAATGGTGTTTGCTAGCTG-3′, and pCMV-luc-miR-124 with primers 5′-CAGCTAGCGGCATTCACCGCGTGCCTTAATGGCATTCACCGCGTGCCTTACTCGAGAC-3′ and 5′-GTCTCGAGTAAGGCACGCGGTGAATGCCATTAAGGCACGCGGTGAATGCCGCTAGCTG-3′. For reporter library construction, total RNA was isolated from 293T, HeLa, and Huh7 cells using Trizol (Invitrogen), mixed, and reverse-transcribed to produce cDNA using a first-strand cDNA synthesis kit (Invitrogen), which was then used as the template in PCR reactions. PCR primers were designed to carry NheI or XhoI sites at their ends for subsequent cloning, and targets with sequences containing internal NheI or XhoI sites were excluded. Afterwards, reactions were run on an agarose gel to confirm that the expected products, typically of ~500 base pairs, were amplified. The DNA was then gel-extracted, digested with NheI and XhoI, ligated to pCMV-luc by T4 DNA Ligase, and transformed into chemically competent Mach1 cells (Invitrogen). Positive clones were screened and identified by PCR, and DNA was isolated using a miniprep kit (Sangon). To check the quality of library preparation, we randomly sequenced approximately 15% of the constructs (Sangon), and insert identities were all verified. Unconfirmed but incorrect inserts might affect the results. To test the importance of MREs in target repression, the Quikchange method (Stratagene, La Jolla, CA, USA) was used to make deletions in the seed regions of select targets, which were then confirmed by DNA sequencing.

### 2.4. Transfection and Reporter Assay

Cells in 24-well plates (Sangon) were transfected using Lipofectamine 2000 (Invitrogen). 293T cells were transfected with approximately 1 ng of a reporter plasmid; 1 ng of pRL-CMV; and 500 ng (unless indicated otherwise) of pSuper-7T, pSuper-gmiR-1, pSuper-gmiR-122, or pSuper-gmiR-124. HeLa cells were transfected with approximately 50 ng of a reporter plasmid; 10 ng of pRL-CMV; and 500 ng of pSuper-7T, pSuper-gmiR-1, pSuper-gmiR-122, or pSuper-gmiR-124. Two days later, cells were lyzed, and firefly and Renilla luciferase activities were measured using the Dual-Luciferase Assay System (Promega). The relative expression of a target reporter (R0) was defined as the ratio of firefly luciferase activity to Renilla luciferase activity under the pSuper-gmiR-1, pSuper-gmiR-122, or pSuper-gmiR-124 transfection condition, divided by the ratio under the negative control, pSuper-7T transfection condition. To further aid in the comparison of the over 190 potential targets per miRNA and minimize the variations due to performing experiments and examining different targets on separate days, a positive control reporter plasmid (pCMV-luc-miR-1, pCMV-luc-miR-122, or pCMV-luc-miR-124) was included in every experiment, and a normalized relative reporter expression (R1) was calculated as the R0 of a target reporter divided by the R0 of the positive control. R0 values of the positive controls were generally between 0.1 and 0.2, whereas those of the target reporters were between 0.5 to 0.95, so their R1 values had a typical range of 3 to 7.

### 2.5. Bioinformatics and Statistical Analyses

RNAFold (http://rna.tbi.univie.ac.at/cgi-bin/RNAWebSuite/RNAfold.cgi) was used to estimate the predicted secondary structural stability (∆G) of 3′ UTR target sequences [41]. AU% values of mRNA sequences were computed using Excel. RNAhybrid (https://bibiserv.cebitec.uni-bielefeld.de/rnahybrid) was used to calculate the hybridization energy (∆G) between miRNAs and target MREs [42]. Select mRNA expression datasets were downloaded from the Gene Expression Omnibus (https://www.ncbi.nlm.nih.gov/geo), with preferences given to the more recently published or performed RNA-seq datasets. The Galaxy website (https://usegalaxy.org) was used to join target gene values with various parameters or expression data of the target genes into single files for correlation studies [43]. SPSS 17 (IBM, Armonk, NY, USA) and Graphpad Prism 7 (GraphPad Software, La Jolla, CA, USA) were used for statistical and Pearson correlation studies. Two-sided Student’s t-tests were performed to compare different target genes, and a one-sided K-S test to compare the influences of different seed types on target repression, with *p* < 0.05 considered statistically significant.

### 2.6. miRNA Overexpression and RNA-seq

293T cells in a six-well plate were transfected with a control RNA, miR-1, miR-122, or miR-124 mimic (Sangon) at 2 μg per well, and 36 h later, total RNA was isolated using Trizol and subjected to RNA-seq (SuperBio, Nanjing, Jiangsu, China). Briefly, a cDNA library was prepared from 1 μg of total RNA using the VAHTS mRNA-seq v2 Library Prep Kit (Illumina, San Diego, CA, USA). HiSeq single-end sequencing was carried out according to standard Illumina protocols. Basecalls were performed using CASAVA version 1.8 (Illumina). Sequenced reads were trimmed for adaptor sequences and masked for low-complexity or low-quality sequences and then mapped to the human GRCh38 genome assembly using hisat2-2.0.0, and fragments per kilobase per million reads were calculated [44,45]. Genes with reads per kilobase per million above 0.004 were considered reliably detected and used in further analyses. RNA-seq data have been deposited in the Gene Expression Omnibus under the accession number GSE123311.

## 3. Results

### 3.1. Construction of Target Reporter Gene Libraries

miR-1, miR-122, and miR-124 targets predicted by TargetScan, miRanda, or PicTar were chosen for reporter library construction, according to Materials and Methods. We then designed primers to perform PCR and clone a segment of the target 3′ UTR centering around the MRE(s) into a firefly luciferase reporter plasmid. Table 1 lists the numbers of reporter genes selected from the three programs. There were a total of 196 putative targets for miR-1 and miR-124, and 194 for miR-122. Because many targets were predicted by two or even three programs, and PicTar predicts fewer targets than TargetScan and miRanda, most targets were found in TargetScan and miRanda predictions (Table 1). The libraries also included previously reported miRNA targets categorized by miRTarBase [27] as potential, positive controls (Table 1). Most of our targets had a single MRE and ~500 nt sequences, in order to maintain consistency for subsequent comparisons. Approximately half of all predicted genes by the three programs contain a single MRE. Our targets had the full range of ranking scores predicted by the programs, as shown in Figure 1.

The left column lists the miRNA target prediction programs and features of the selections. For example, we tested a total of 196 genes for miR-1: 165 were predicted by TargetScan 7.0, 159 by miRanda, 84 by PicTar, 19 by TargetScan alone, 28 by miRanda alone, 1 by PicTar alone, and 64 by all three programs. Additionally, 10 were previously reported according to miRTarBase, and 171 of the targets contained a single MRE, while 25 contained two or more MREs.

### 3.2. Reporter Assays to Measure Target Repression by miRNAs

We next transiently transfected luciferase constructs with or without miRNA-expressing plasmids into 293T cells. Human 293T cells lack miR-1, miR-122, and miR-124, so luciferase expression could be considered free of interference from these endogenous miRNAs and their repressive activities and hence, has a high signal-to-noise ratio. Preliminary studies showed a dose-dependent increase in the degree of reporter inhibition by miRNA-expressing plasmids, although inhibition plateaued quickly (Appendix A). To ensure we identified as many targets as possible and also the maximal inhibitory effects by miRNAs, we thus used 500 ng of miRNA-expressing plasmids in subsequent transfections in 24-well plates. Relative reporter expression values, R0 and R1, were then calculated (Materials and Methods). The larger the R0 and R1 values, the weaker the inhibition by a miRNA. Every reporter construct was tested in at least three biological replicates. If R0 of a reporter was less than 1, then the corresponding mRNA was considered a miRNA target; otherwise, the online prediction was deemed incorrect. As presented in Figure 2, a minority of reporters yielded R0 ≥ 1, indicating that they were unlikely miRNA targets. Most reporters, however, were inhibited by miRNAs, and their R0 values typically varied from ~0.5 to 0.95 (R0 < 1, with Student’s *t*-test, *p* < 0.05; Figure 2). R1 values were also calculated to help compare targets examined in different experiments, and they similarly varied (Figure 2). Summaries of target confirmation are provided in Table 1, additional data are shown in Appendix A, and all the genes tested in our reporter assays are listed in Appendix A.

To demonstrate the specificity of miRNA action, we deleted mRNA sequences complementary to the miRNA seed region [2] in select confirmed targets for miR-1, miR-122, and miR-124, and then compared the responses of wildtype (WT) and mutant (MUT) reporters to cognate miRNAs. Representative results are shown in Figure 3: MUT reporters had higher R0 values than the WT, indicating the dependence of miRNA inhibition on MREs. In addition, we randomly chose miR-1 and miR-124 reporter constructs and transiently transfected them into HeLa cells. Their R0 values and ranking closely matched those obtained in 293T cells (Figure 4). Together, our results directly verified hundreds of miRNA targets, most of which had not been previously published or rigorously tested, and demonstrated that miRNAs varied in the suppression of different targets.

### 3.3. Parameters that Correlated with Target Repression by miRNAs

Through reporter assays, we obtained data on direct miRNA inhibition for a large number of target mRNAs. The next question is, what factors determine whether an mRNA is a miRNA target or not, as well as how much a target is repressed by a miRNA? We thus performed correlation and statistical analyses, based on the R0 and R1 values of reporter genes. R0 and R1 yielded qualitatively similar results, so only R1 data are shown here.

We found little difference in the confirmation rates of TargetScan, miRanda, and PicTar predictions, ranging from approximately 72 to 85% (Table 1). Targets predicted by all three programs or known targets (miRTarBase) were also well-confirmed, albeit with a smaller sample size (Table 1). Few parameters significantly or consistently differentiated between confirmed and unconfirmed targets; these included program ranking scores (Appendix A), and AU% and the predicted secondary structure stability (ΔG) of the target sequences (Appendix A). Although confirmed targets tended to have higher absolute values of the scores, few comparisons reached the level of significance (Appendix A). Examining more predictions might help, or experimental approaches may be the only way to screen for genuine targets in the predictions. On the other hand, the ranking scores of TargetScan correlated modestly and significantly with how well targets were repressed by all three miRNAs (r = 0.237–0.391, *p* < 0.05; Figure 5): the more likely a miRNA target was, according to TargetScan, the stronger the corresponding reporter was repressed. miRanda scores had lower correlation coefficients, while PicTar scores were not predictive (Figure 5).

TargetScan and PicTar mandated target site conservation [23,25]. Most of our miRanda predictions are also shared by TargetScan and/or PicTar (Table 1), and it is difficult to rule out sequence conservation. Therefore, a vast majority of our targets, if not all, are deemed conserved. Nevertheless, TargetScan 7.0 predictions included conserved sites and poorly conserved sites. If we compared those two categories (Appendix A), we found that conserved sites were slightly better confirmed as miR-1 and miR-122 targets than poorly conserved sites. For miR-124, the pattern was reversed, but the poorly conserved sites had a much smaller sample size (Appendix A). Overall, all groups were well-verified, with only minor differences (Table 1 and Appendix A). The R1 values for conserved sites were lower than those of poorly conserved sites (Appendix A; for miR-122 and miR-124 targets, *p* < 0.05), indicating that the latter were inhibited more weakly. These results suggested that more conserved targets might be better selected and inhibited by miRNAs.

Newer algorithms have been developed to predict miRNA targets or integrate multiple programs, such as mirDIP [46]. We examined how mirDIP 4.1 performed, using the top 1% score class (Appendix A). Targets were confirmed at rates (72–86%) similar to those in Table 1, suggesting little improvement in terms of confirmation. Confirmed mirDIP targets, however, had statistically lower R1 values than those confirmed, yet not predicted, by mirDIP (Appendix A).

The canonical seed region (nt 2–8) and its variations have been shown to dominate miRNA function [2]. Indeed, we found that the more extensive pairing at the seed, the stronger the inhibition, for all three miRNAs (Figure 6). For miR-1 targets, 8mer sites were more repressed than 7mer-m8 (*p* = 0.001) and no seed match sites (denoted as “else” in Figure 6; *p* = 0.005). For miR-122 targets, 8mer sites were more repressed than 6mer (*p* = 0.020) and no seed match sites (*p* = 0.003), and 7mer-m8 sites more repressed than no seed match sites (*p* = 0.005). For miR-124 targets, 8mer sites were more repressed than 7mer-A1 sites (*p* = 0.026). We did not always detect differences at a significant level between other seed groups, most likely due to smaller sample sizes. Targets with a single MRE, as a group, were repressed slightly less well than those with two or more MREs, but the differences were not significant (Appendix A). Explanation of this result must take into account how we selected our original targets: fewer than 20% contained more than one MRE, whereas approximately 50% of all the targets predicted by the three programs had more than one MRE.

The profiling strategy has identified local target RNA features as important determinants of miRNA functions [9,12]. We found that the AU% and ΔG of the predicted secondary structure of cloned target 3′UTR correlated weakly, but significantly, with the degree of reporter inhibition by miR-1 and miR-122, but not by miR-124: the more flexible the mRNA, the more inhibition exhibited by miR-1 and miR-122 (r = ~0.2–0.3, *p* < 0.05, Table 2). We also calculated the local AU content according to Grimson et al. [9] and found that it correlated negatively with R1 for all three miRNAs, as predicted, but only miR-122 targets achieved a significant correlation (Table 2). Lastly, the hybridization energy between miR-1 (but not miR-122 and miR-124) and MREs positively correlated with target R1 values, suggesting that tighter binding contributed to stronger inhibition by miR-1 (Table 2). Because most of our reporters contain a “standardized”, ~500 nt 3′UTR segment, we were not able to evaluate other information, such as MRE locations in endogenous mRNAs and open reading frame length [12].

The sample size (*n*), correlation coefficients (r), and *p*-values are listed, with those with *p* < 0.05 marked in red. AU% calculated the percentages of AU residues in the target mRNAs centering around the seed matches. For example, 50 nt included 25 nt upstream and 25 nt downstream of the seed matches in single MRE-containing targets. Secondary structures were likewise predicted using the same sequences. Exceptions were those marked with “whole” in the table, which included the full length mRNA inserts (~500 nt). The ΔG of miRNA and MRE hybrid was predicted using RNAhybrid. Local AU content was calculated for RNAs containing the four seed types [9].

### 3.4. Relationships between Reporter Inhibition and Endogenous Gene Expression

So far we have confirmed miRNA target genes using reporter assays. Nevertheless, is the expression of endogenous human genes also inhibited by miRNAs in a similar fashion? That is, if miRNAs contribute to differential gene expression in vivo, one might expect the lower the R1 value, the lower the endogenous gene is expressed. Because miRNA action usually leads to target mRNA reduction [2], we examined publicly available RNA expression data in human heart, liver, and neuronal tissues and cells and considered only genes that were detected in the samples. For miR-1 and miR-122, positive but non-significant correlations were observed between the R1 values of confirmed targets and their corresponding mRNA expression in human heart and liver tissues [47], respectively (Figure 7A,B). For miR-124, however, weak and positive but significant correlations were found in the spinal cord, brain, and neural cells (Figure 7C). If we compared the R1 values of miR-124 targets to the corresponding gene expression in the heart and liver tissues, there were no significant correlations. For example, with the human heart mRNA expression dataset GSM2072383, we obtained *n* = 97, r = −0.016, and *p* = 0.878, and with GSM2453454, we obtained *n* = 88, r = −0.01, and *p* = 0.927. These results, therefore, further validated the specificity of our analyses.

### 3.5. Differential Reduction of Target mRNA Expression by miRNAs

Lastly, we examined how the expression of endogenous target genes was affected by miRNA overexpression in human cells. We transiently transfected miR-1, miR-122, or miR-124 mimic into 293T cells, determined the global mRNA expression by RNA-seq, and then calculated how the confirmed targets changed their mRNA expression. Results of all the target genes, regardless of whether their levels went down or not upon miRNA transfection, are presented in Figure 8. For miR-124 targets, there was no significant correlation between R1 and the degrees of mRNA expression changes (Figure 8). For miR-1 and miR-122, however, there was a positive and significant correlation (Figure 8), suggesting that the lower the R1 value, the larger the reduction of target mRNA levels in cells, consistent with the idea that differential miRNA suppression influences endogenous gene expression.

## 4. Discussion

Dozens of miRNA target prediction platforms have been developed, and a great deal of research has aimed at identifying miRNA target genes. Despite all these efforts, however, the performance of the prediction programs has rarely been evaluated experimentally, and most miRNA targets have not been tested rigorously. This study represents the first extensive comparison of three venerable programs, TargetScan, miRanda, and PicTar, by directly examining a large number of predicted target genes using a standard luciferase reporter assay. The three programs behaved similarly, with 72–85% success rates (Table 1). For comparison, earlier luciferase assays verified 11 out of 15 and 6 out of 7 predicted TargetScan and PicTar targets, respectively [23,25]. Of course, whether these genes are truly miRNA targets in vivo requires further analyses. As indicated by Table 1, most of our selections were shared by two programs, and almost all PicTar ones were also predicted by TargetScan or miRanda, even though we did not deliberately choose these overlapping predictions. This observation reveals extensive overlap in predictions among the programs. RNA base pairing is the most critical factor of consideration in all prediction algorithms, so perhaps it is natural that different programs performed similarly. With different programs predicting many common, as well as different, genes, it is likely that there are altogether thousands of genuine target genes for the three miRNAs we tested [37]. Interestingly, target ranking scores provided by the programs did not consistently predict whether a candidate target is likely to be confirmed or not (Appendix A). A caveat, however, is that our libraries were not an exact snapshot of program predictions, as most of our selections are single MRE genes, whereas only approximately 50% of all predictions have a single MRE. Additionally, because we aimed to cover the whole range of ranking scores, some scores may be under-represented in terms of their gene numbers (Figure 1).

How distinct targets may be inhibited by the same miRNA differentially has been overlooked. Previous studies examined only a dozen or so native sequences using reporter assays, or the general efficacy of different seed groups using the profiling strategy, which was not able to distinguish between direct and indirect effects [9,12,21,23,24,25,29,31,48]. Our results systematically demonstrated that the miRNA inhibition of individual targets varies in extent (Figure 2 and Appendix A). Targets with MREs that are more conserved might be better selected and inhibited by miRNAs (Appendix A). Furthermore, of the three programs, the TargetScan scores best predict how strongly a target might be inhibited by a miRNA, while PicTar scores are not informative (Figure 5). This is likely because, being the most recently updated, TargetScan has incorporated more features trained on experimental datasets concerning the efficacy of mRNA suppression [12]. Because most of our tested genes were predicted by two programs, and almost all selected PicTar targets were also predicted by TargetScan or miRanda (Table 1), this result suggests that while PicTar predicts miRNA targets with a comparable accuracy, its scoring system lacks information on how well a target is inhibited by a miRNA. Therefore, in a sense, being targeted by a miRNA and how well it is being targeted remain two separate issues. Newer programs such as mirDIP did not have significantly enhanced target confirmation ratios (Appendix A), likely because prior algorithms already performed proficiently in this regard (Table 1). Having incorporated information provided by programs like TargetScan (Figure 5), however, mirDIP might be better equipped to discriminate between good and poor miRNA substrates (Appendix A).

With reporter assay results, we then examined how various mRNA features, previously considered critical based on studies employing the indirect, profiling strategy, affected differential gene suppression by miRNAs. We confirmed, from a different angle, that the more extensive base pairing at the seed region, the stronger the inhibition by miRNAs (Figure 6). We also found a general trend of a higher local target AU content and lower secondary structure stability favoring miRNA repression, which was true for miR-1 and miR-122, but not for miR-124 (Table 2). In addition, tighter binding between miR-1 and MRE correlated with stronger inhibition, although the same did not apply to miR-122 or miR-124 (Table 2). Slutskin et al. [26] identified a much stronger contribution by MRE to repression by a different set of miRNAs, but they examined rationally designed sequences, which are more homogenous than native mRNAs, and other studies did not reveal such a dominant effect by MRE [9,12]. On the other hand, the reason behind discrepancies in our results for miR-1, miR-122, and miR-124 is unclear. While it has been assumed that the same mechanisms apply to all miRNAs, there have been indications that miRNAs such as miR-124 might have additional binding preferences for certain mRNA features [14]. Slutskin et al. [26] have also reported miRNA-specific effects on gene repression. Moreover, it was shown that seed-pairing stability contributes to miRNA function, so it is worth noting that the seed-pairing stability of miR-1 is −6.61 kcal/mol, while that of miR-122 and miR-124 is −8.54 and −8.40 kcal/mol, respectively [29].

Highly expressed mRNAs often lack certain MREs, perhaps due to the selection pressure to avoid inhibition by miRNAs [24,49], but does differential miRNA repression also actively regulate endogenous target expression? By analyzing publicly available gene expression datasets, we showed that reporter inhibition by miR-124 correlated statistically significantly with endogenous gene expression; i.e., the stronger a reporter gene is repressed, the weaker the corresponding gene is expressed in the human nervous system and neural cells (Figure 7C). We did not obtain significant correlations with miR-1 and miR-122 (Figure 7A,B); future work analyzing more miRNAs and datasets is needed to understand whether miRNAs also differ in their global, regulatory effects on endogenous gene expression. As for the weak correlations shown in Figure 7, there are several explanations. First of all, gene expression is controlled at multiple levels. Transcription is often considered the principal determinant, and specific proteins can play a dominant role in individual mRNAs’ expression, while miRNAs are believed to fine-tune expression [2]. It is, therefore, not surprising that miRNA action might play only a minor role. Additional complications include the same genes being targeted by other miRNAs and producing various mRNA isoforms and 3′ UTRs with or without MREs. Our data were also skewed toward genes with a single MRE, whereas those with multiple MREs could be more susceptible to miRNA action. Lastly, reporter suppression by miRNAs was acquired in an artificial setting and generally weak, between 10 and 50%, which makes precise determination of the inhibitory effects difficult and data noisy (Figure 2 and Appendix A). Noise obviously also exists in gene expression datasets, all leading to the underestimation of correlation coefficients [50]. Nevertheless, our analyses are highly specific, as the same miR-124 targets are not differentially repressed, according to their R1 values by miR-124, in human heart and liver tissues that lack miR-124. Therefore, our results point to a new mechanism where miRNA differential inhibition of target genes regulates endogenous gene expression. Moreover, a miRNA is usually outnumbered by its potential binding sites in vivo [12,19,47,51], so a better substrate may be more likely to be targeted by a miRNA. Consequently, even though our reporter assays suggested thousands of targets for a miRNA, it remains probable that inside a cell, only a fraction are effectively inhibited by the miRNA, and to different extents.

In conclusion, we rigorously confirmed hundreds of target genes for human miR-1, miR-122, and miR-124, laying a foundation for further studying their functions and regulation and the mechanisms of action by the miRNAs. We showed that miRNAs repress the expression of different target genes to varying degrees and identified parameters that can predict or explain this phenomenon, which we also suggest contributes to differential gene expression in vivo. While previous studies had focused on determining whether a gene is a target of a miRNA or not and aggregated genomics profiling data of different miRNAs to identify common mechanisms by miRNAs, we propose that it is worth treating such a large number of targets, as well as miRNAs, differently.

## Figures and Tables

**Figure 1 cells-08-00791-f001:**
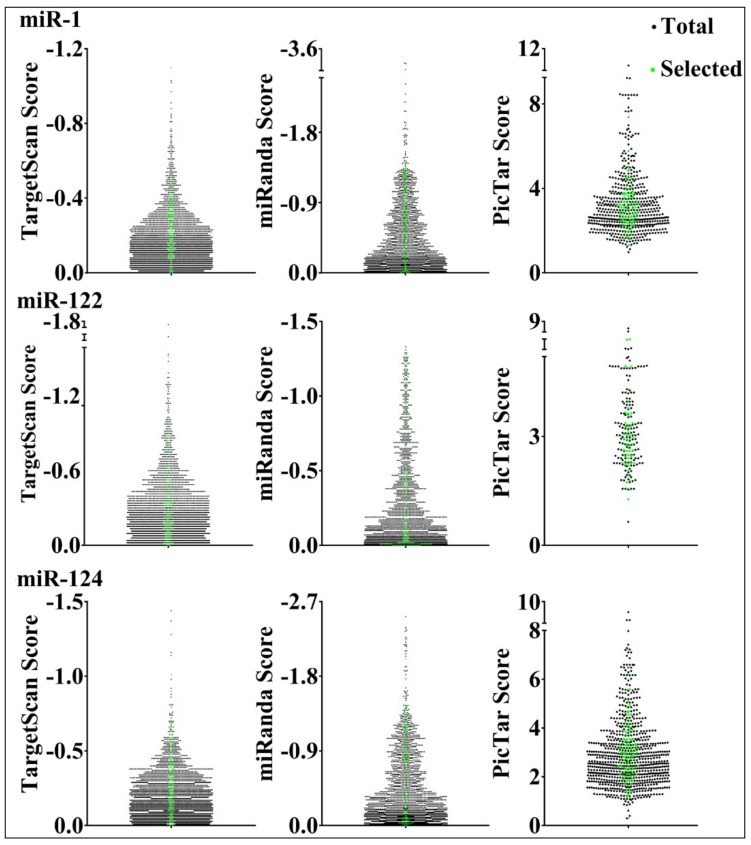
Selection of predicted miRNA target genes. Ranking scores provided by TargetScan, miRanda, and PicTar are shown in the *y*-axes. Black dots denote all the predicted genes, and green dots the ones chosen for testing.

**Figure 2 cells-08-00791-f002:**
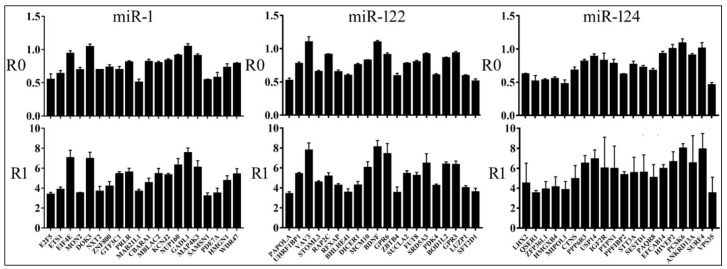
Results of reporter assays in 293T cells. Averages and standard deviations of R0 and R1 values (*y*-axes) are shown for the individual genes listed at the bottom (*x*-axes).

**Figure 3 cells-08-00791-f003:**
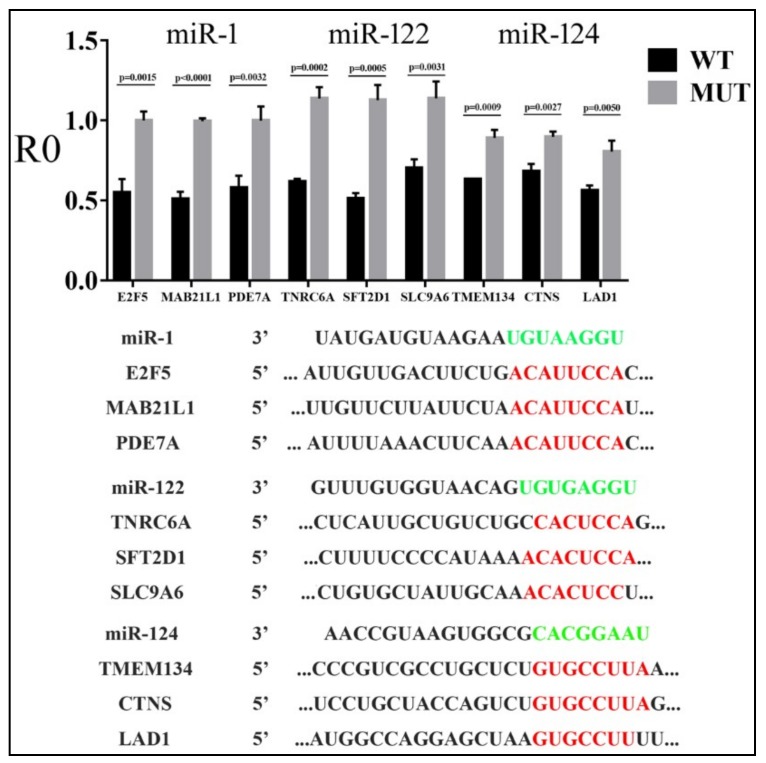
Effects of miRNAs on the expression of wildtype (WT) and mutant (MUT) target reporters in 293T cells. *Y*-axis is the R0 value, and *x*-axis represents individual genes. Averages, standard deviations, and *p*-values between the WT and MUT are indicated on the graph. Sequences of the miRNAs and WT targets are shown below the graph, with miRNA seed regions in green, and the complementary WT sequences deleted in MUT constructs in red.

**Figure 4 cells-08-00791-f004:**
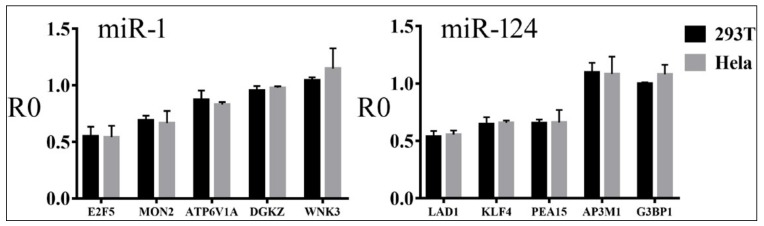
Reporter assay results in 293T cells and HeLa cells. *Y*-axes are the R0 values, and *x*-axes are the potential miR-1 and miR-124 targets. Averages and standard deviations are shown.

**Figure 5 cells-08-00791-f005:**
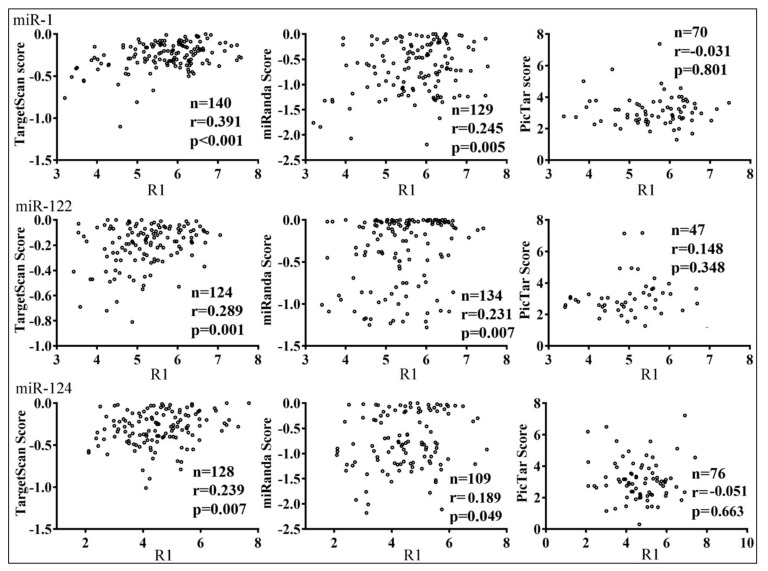
Correlations between miRNA reporter inhibition indexes (R1, *x*-axes) and the scores of TargetScan, miRanda, and PicTar (*y*-axes). Little circles represent individual confirmed targets. The sample size (*n*), correlation coefficients (r), and *p*-values are listed.

**Figure 6 cells-08-00791-f006:**
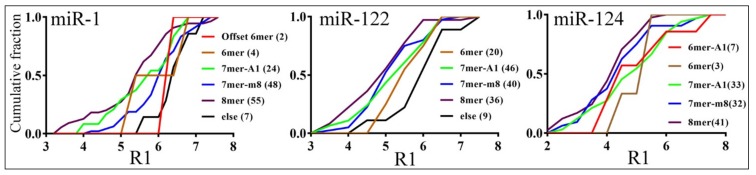
Effects of seed types on the degrees of reporter inhibition by miRNAs. *X*-axes show the R1 values, and *y*-axes the cumulative fraction. Color lines represent different seed types in the confirmed, single MRE-containing targets, with each sample size in parentheses. Seed types were categorized according to Bartel [2], and “else” contained targets not belonging to all the other types.

**Figure 7 cells-08-00791-f007:**
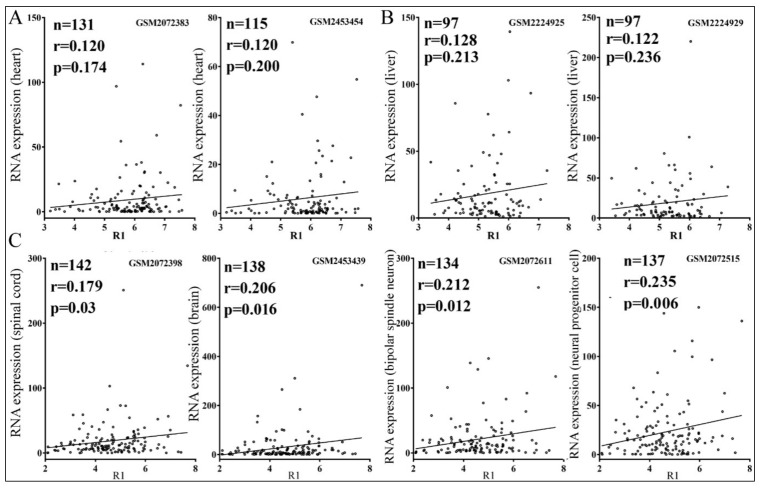
Pearson correlations between human endogenous gene expression and target reporter inhibition. (**A**) Correlations between RNA expression in the heart (*y*-axes) and miR-1 target R1 values (*x*-axes). (**B**) Correlations between RNA expression in the liver (*y*-axes) and miR-122 target R1 values (*x*-axes). (**C**) Correlations between RNA expression in the nervous system and neurons (*y*-axes) and miR-124 target R1 values (*x*-axes). Little circles represent individual confirmed targets. The sample size (*n*), correlation coefficients (r), *p*-values, and the corresponding Gene Expression Omnibus dataset accession numbers are indicated.

**Figure 8 cells-08-00791-f008:**
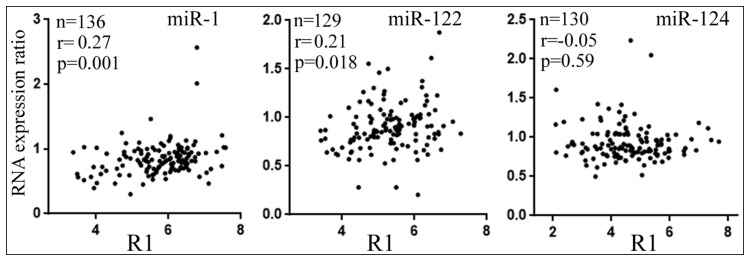
Correlations between R1 values and changes in target mRNA levels upon miRNA transfection in 293T cells. *Y*-axes show RNA levels under miRNA transfection condition divided by those under the control RNA transfection condition. Dots represent individual confirmed targets. The sample size (*n*), correlation coefficients (r), and *p*-values are indicated.

**Table 1 cells-08-00791-t001:** Summaries of miRNA target confirmation results.

	miR-1			miR-122			miR-124		
	Selected	Confirmed	Ratio	Selected	Confirmed	Ratio	Selected	Confirmed	Ratio
TargetScan	165	140	84.8%	149	124	83.2%	174	128	73.6%
miRanda	159	129	81.1%	162	134	82.7%	151	109	72.2%
PicTar	84	70	83.3%	57	47	82.5%	104	76	73.1%
TargetScan only	19	17	89.5%	19	17	89.5%	13	11	84.6%
miRanda only	28	19	67.9%	42	34	84.0%	9	8	88.9%
PicTar only	1	1	100%	2	2	100%	8	6	75.0%
All three	64	52	81.3%	43	33	76.7%	67	48	71.6%
miRTarBase	10	10	100%	14	13	92.9%	5	5	100%
Single MRE	171	140	81.9%	181	152	84.0%	166	117	70.5%
Multiple MREs	25	22	88.0%	13	11	84.6%	30	28	93.3%
Total	196	162	82.7%	194	163	84.0%	196	145	74.0%

**Table 2 cells-08-00791-t002:** Correlations between R1 values and mRNA properties.

mRNA Properties		miR-1			miR-122			miR-124		
		*n*	r	*p*	*n*	r	*p*	*n*	r	*p*
**AU%**	50 nt	131	−0.25	0.004	143	−0.32	<0.001	116	0.00	0.991
	100 nt	130	−0.22	0.013	136	−0.24	0.006	115	0.03	0.716
	200 nt	110	−0.23	0.017	125	−0.25	0.005	99	−0.07	0.502
	300 nt	84	−0.17	0.120	98	−0.24	0.020	77	−0.04	0.713
	whole	162	−0.17	0.032	163	−0.15	0.061	145	0.05	0.583
ΔG of predicted secondary structure	50 nt	131	−0.11	0.198	143	−0.22	0.009	116	−0.01	0.899
	100 nt	130	−0.18	0.046	136	−0.23	0.008	115	0.02	0.843
	200 nt	110	−0.20	0.033	125	−0.27	0.003	99	−0.03	0.751
	300 nt	84	−0.13	0.250	98	−0.27	0.008	77	−0.04	0.711
	whole	162	−0.18	0.024	163	−0.03	0.685	145	−0.09	0.276
ΔG of miRNA and MRE hybrid		140	0.23	0.005	152	−0.01	0.926	117	0.03	0.722
Local AU content		129	−0.14	0.109	140	−0.32	<0.001	107	−0.12	0.219

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
