# Peer review of "Differential Inhibition of Target Gene Expression by Human microRNAs"

_cells, 2019, doi:10.3390/cells8080791_

Round 1
Reviewer 1 Report
In this work, Li and colleagues investigate the targeting of close to 200 reporters for miR-1, miR-122, and miR-124 (close to 600 in total). This systematic analysis of a large dataset of targets of these miRNAs allows them to compare the efficacy of the predictions by Targetscan, miranda, and Pictar. The large set of targets used also allowed the authors to look at sequence properties associating with increased down-regulation of the targets - which for instance confirmed previous reports that the length of the seed interaction was associated with more repression of the targets. The authors also investigate the relationship of the reporter repression with that of endogenous mRNAs upon miRNA over expression - showing that for miR-1 and miR-122 better targets in the reporter assays there was a correlation with stronger repression of endogenous mRNAs.
Altogether, this manuscript is very well written and the approach of cloning so many vectors is significant to improve our understanding of miRNA targeting. Nonetheless, this reviewer feels a few additional analyses should be carried out to really make the most of their unique dataset and truely advance the field.
Major points:
1- The methods are well presented but a few things were unclear to understand how the library was generated. First, were the target sites chosen conserved between species? (this is an important point since interspecies conservation is applied by targetscan, and may have also been used in miranda - the link used points to microrna.org which gives the possibility to get miranda targets with different degree of targeting).
As such, when the authors present the scores of the software used, did they include conserved and non conserved targets among their pools of targets? It would be important to see how conserved vs non conserved sites perform for efficiency of targeting, and repression strength. In addition, miranda, with low conservation parameters (and relaxed DeltaG), can basically predicts sites for as many as >10,000 targets. Is it possible that although selected to avoid multiple target sites (presumably conserved sites), some of the targets in fact have many poorly conserved sites but still biologically relevant?
In addition, the way the cloning of the vectors was performed is not entirely clear. Presumably, NheI and XhoI sites were added to the ends of the primers used to amplify each target? Does this mean the targets were selected to avoid these two sites (which otherwise would have cut the amplicon?). The terminology library is misleading as it usually refers to a preparation of pools of vectors in the same ligation - but presumably here each vector was cloned individually? (and mini-prepped individually?). This should be detailed (which ligation, bacteria were used, transformation technique...). Finally, one of the key points of this paper is the generation of these vectors, but the authors only validated 15% of them (i.e. 90). It seems that the authors could easily sequence these vectors pooled through next gen sequencing - but ALL the vectors should be validated.
2- One of the arguments of the authors for their study is that the use of RNAseq approaches (for instance based on Ago2 pull-down) does not fully confirm direct binding of the miRNA to its target (depending on the approach used), and binding does not equate repression. Since the miRTarbase targets used here were selected for reporter validation they did not include the targets which have been derived from Ago2 pull down or more recent miRNA:mRNA capture techniques such as CLASH or CLEAR. The authors should look at the targets they investigated and see how many were captured in miRTarbase including these high throughput approaches - and see whether, for instance, or many failed to be validated. It would also be important to consider the impact of non-canonical sites (validated in high throughput miRTarbase targets), for the targets which they investigate and whether these sites could have a compounded effect on targeting - explaining increased repression.
3- Is there an overlap between some of the targets used for the 3 miRNAs? In other word, how many targets are shared? If there are a few, it would be worthwhile comparing the efficacy of target repression between the miRNAs - to assess whether some 3UTRs are just more amenable than others to miRNA repression (for instance due to simpler secondary structures).
4- Aside from targetscan, miranda and Pictar are very dated algorithms. The authors propose that the accuracy of these algorithms compares well, and that concurrent predicted targeting does not really improve the efficacy of the prediction. It would be important to make use of more recent softwares, for instance miRDip (which integrates many prediction softwares) to see whether accuracy of concurrent predictions from more recent tools can perform better to pinpoint the true targets.
Minor point:
It seems that this resource of vectors should be made available to the scientific community - for instance through AddGene.
Author Response
Thanks for the detailed comments. Please see the attachment.

Reviewer 2 Report
The study from Li and colleagues entitled "Differential inhibition of target gene expression by human microRNAs" aimed to investigate the reliability of miRNA target predictions of three miRNAs: miR-1, -122 and - 124. While it is impressive the amount of work put on this manuscript, the scientific merit is not robust. There are studies that have already evaluated the success rates of target predictions, and while it was conducted on a small scale, in my opinion doing in hundreds is only an extension of what is already published. Another point that is no clear to me is the RNA-seq strategy, the method is completely missing and its importance for this study was not clear.
Author Response
As acknowledged in the manuscript, our confirmation rates are comparable to previous estimates. But some of the new findings, with stronger evidence, include:
1. Program scores do not significantly discriminate between targets as non-targets.
2. Different targets are inhibited to different degrees.
3. While all three programs predict targets with similar success rates, performance of their scores varies with respective to how well target genes are inhibited, suggesting that predicting target genes and predicting gene repression levels remain separate problems. This is true even with the more updated programs like mirDIP (new Figure S6).
4. We analyzed how various parameters could correlate to or explain differential miRNA repression. All previous work was done using mRNA-profiling data, so we feel it is refreshing to apply data from a different system and angle, and to compare the results.
5. Comparing the results, we found that we could corroborate some but not all of the previous conclusions. Clearly a number of explanations exist, and future studies are needed. But one interesting aspect to point out is that gene-profiling analyses often treated distinct miRNAs as the same and evaluated their aggregate results (e.g., refs 9, 12), but our data suggested that, analogous to target genes not being the same, miRNAs might also be different.
6. We correlated miRNA repression data to endogenous mRNA expression data and found that miR-124 activity might contribute to differential endogenous gene expression in the nervous system (Figure 7). This feature had not been reported or suggested for any miRNA. It is specific since miR-124 activity does not impact gene expression in the heart or liver.
7. Considering thousands of targets are predicted, and the confirmation rates are 70-80%, targets in vivo may well number in the thousands. But as reported (refs 12,19,47,51) and discussed in the manuscript, many factors influence miRNA-mRNA interactions in vivo, and a new, critical consideration should be the differential activities of miRNAs, largely overlooked before.
RNA-seq methodology has been updated in purple in the revised manuscript, and two more references [44,45] added. The sequencing and data analysis parts were done mostly by the company, and the information had been provided in our submission to GEO at NCBI. Our goal here was to simulate a situation when miRNA levels change in cells, target mRNA levels also change, but unevenly, correlating to (reflecting) differential repression by miRNAs. For our actual data with miR-1 and miR-122, p <0.05 (Figure 8).
Reviewer 3 Report
This manuscript describes that systematical investigation of miRNA target genes with miR-1, miR-122, and miR-124 showed that a mechanism by which differential target repression by miRNAs regulates endogenous gene expression. Although the results are limited with 3 miRNAs, this is an interesting study.
You need to explain more the reasons why you chose these 3 miRNAs.
You need to indicate the numbers of fold increases of miRNAs in Figure S1.
It is ideal to also study using siRNAs.
Author Response
The manuscript has indicated that miR-1, miR-122, and miR-124 are three relatively well-known, tissue-specific miRNAs that have been shown to have important biological functions (refs. 32-36). It has also compared our results with some of the prior data and conclusions drawn from studying those miRNAs (refs. 9,12,14,19,21,24,25,29,31). And that the cell culture systems we used, 293T and HeLa cells, lack these miRNAs, so our luciferase reporter assays would be free of interference from endogenous miRNAs (and their repressive activities) and hence, have a high signal-to-noise ratio. Additional reasons, some added to the revised Introduction in purple, are that we’d then be able to examine endogenous mRNA expression in tissues such as the heart, liver, brain, and correlate it to target inhibition by specific miRNAs (Figure 7), and that in the future we’d compare our verified target genes with published data and further study the functions of these genes in the respective tissues or organs (e.g., refs 14,21,24,32-36).
Figure S1 has been updated with qPCR results of the overexpressed miRNAs. qPCR methods with a reference are provided.
siRNAs and miRNAs work by the same mechanisms (please also see Zeng et al., PNAS, 2003, 100: 9779-9784), and we have used both commercial miRNA mimics (like siRNAs) and miRNA-encoding plasmids to inhibit target genes. Results were similar, as generally expected and reported in the literature. Here we chose to use plasmids in reporter assays simply because miRNA mimics are more expensive, less stable, and harder to quantify. But in experiments looking at the repression of endogenous genes, we use miRNA mimics.
Round 2
Reviewer 1 Report
The authors have addressed my comments, and the revised draft is improved. This reviewer understands that High throughput sequencing will not cover the entire length of the inserts, but it would however rapidly confirm that at least the correct regions were cloned. The fact that all the tested vectors were validated by sanger sequencing of the 15% vectors is promising, but cannot really be used to confirm the broad accuracy of the method. This reviewer still feels more should be done to confirm the correct regions were cloned - by either sequencing of some sort or using PCRs with internal primers (targeted to the sequence within the cloned amplicon).
Author Response
We agree that sequencing all the plasmids is the ultimate way to confirm 100% cloning success, but this is rather costly and time-consuming. This problem is common in statistics and dealt with by attribute sampling, acceptance testing, quality control, etc. In our case, assuming no clone is allowed incorrect, n, denoting test sample size, = ln(1-confidence)/ln(reliability). Thus, if we need to have 95% confidence with 95% reliability, we’d test 59 clones. With n=90 tested all correct, we achieved 99% confidence with 95% reliability, or 95% confidence with 97% reliability. One also needs to consider how we got the clones: we designed specific primers, PCR gave a single, sharp band, product was of expected size, comparing to the prediction as well as to other target gene PCR products, and after cloning we checked the insert by PCR again. So this process ensured additional quality controls. On the other hand, we do still continue sequencing our clones, either when we prepare more DNA from the stocks, or when we need to make mutations. If a construct is indeed requested, we will also sequence the DNA before sending it away.
Reviewer 2 Report
Although the authors have provided a better explanation for their work, in my opinion, this study does not bring novelty to the field. The content was improved and the methods are clear. I have no additional concerns regarding the manuscript.
Author Response
Thanks. It is natural to have different opinions about novelty. One more thing we’d like stress, objectively, is that that in all the literature, few studies (not just in the miRNA field) have research designs, strategies, and results analogous to ours.